# Neuronal NPR-15 modulates molecular and behavioral immune responses via the amphid sensory neuron-intestinal axis in *C. elegans*

**Benson Otarigho[1], Anna Frances Butts[1], Alejandro Aballay[1,2]***

[1]Department of Genetics, The University of Texas MD Anderson Cancer Center, Houston, United States; [2]Department of Microbiology and Molecular Genetics, McGovern Medical School at UTHealth, Houston, United States

*For correspondence:
aaballay@mdanderson.org

**Abstract** The survival of hosts during infections relies on their ability to mount effective molecular and behavioral immune responses. Despite extensive research on these defense strategies in various species, including the model organism *Caenorhabditis elegans*, the neural mechanisms underlying their interaction remain poorly understood. Previous studies have highlighted the role of neural G-protein-coupled receptors (GPCRs) in regulating both immunity and pathogen avoidance, which is particularly dependent on aerotaxis. To address this knowledge gap, we conducted a screen of mutants in neuropeptide receptor family genes. We found that loss-of-function mutations in *npr-15* activated immunity while suppressing pathogen avoidance behavior. Through further analysis, NPR-15 was found to regulate immunity by modulating the activity of key transcription factors, namely GATA/ELT-2 and TFEB/HLH-30. Surprisingly, the lack of pathogen avoidance of *npr-15* mutant animals was not influenced by oxygen levels. Moreover, our studies revealed that the amphid sensory neuron ASJ is involved in mediating the immune and behavioral responses orchestrated by NPR-15. Additionally, NPR-15 was found to regulate avoidance behavior via the TRPM (transient receptor potential melastatin) gene, GON-2, which may sense the intestinal distension caused by bacterial colonization to elicit pathogen avoidance. Our study contributes to a broader understanding of host defense strategies and mechanisms underlining the interaction between molecular and behavioral immune responses.

## eLife assessment

The **important** work by Aballay et al. significantly advances our understanding of how G protein-coupled receptors (GPCRs) regulate immunity and pathogen avoidance. The authors provide **convincing** evidence for the GPCR NPR-15 to mediate immunity by altering the activity of several key transcription factors. This work will be of broad interest to immunologists.

## Introduction

Hosts employ multiple defense mechanisms to combat infections, including molecular immune defenses (*Netea et al., 2019*; *Gourbal et al., 2018*; *Blander and Sander, 2012*) and behavioral defense responses to invading pathogens (*Meisel and Kim, 2014*; *Sarabian et al., 2018*; *Hart and Hart, 2018*). Overall, these strategies are conserved across species (*Sarabian et al., 2018*; *Hart and Hart, 2018*; *Kimbrell and Beutler, 2001*; *Flajnik and Du Pasquier, 2004*), but their relationship and mechanistic interplay are not yet fully elucidated. While the immunological defense response

is effective, it is metabolically costly and may lead to inflammatory damage (*Levine et al., 2011*; *Netea et al., 2020*; *Xiao, 2017*; *Geremia et al., 2014*). On the other hand, the avoidance behavioral response serves as a crucial first line of defense, enabling hosts to prevent or minimize contact with pathogens (*Behringer et al., 2006*; *Curtis, 2014*; *Meisel and Kim, 2014*). Although both immune and behavioral responses to pathogen infection are well documented in *Caenorhabditis elegans* (*Styer et al., 2008*; *Chang et al., 2011*; *Reddy et al., 2011*; *Singh and Aballay, 2019a*; *Sun et al., 2011*), the relationship between these survival strategies remains poorly understood.

*C. elegans* is a valuable model organism for studying the genetic mechanisms that control host immune and behavioral responses to pathogens (*Balla and Troemel, 2013*; *Schulenburg and Félix, 2017*). Although *C. elegans* lacks adaptive immunity, part of its innate immune response comprises evolutionarily conserved pathways and immune effectors (*Schulenburg and Félix, 2017*). *C. elegans* has been widely used in research studies to investigate these pathways due to its well-defined nervous system and genetic tractability, making it an ideal model organism to explore pathways that are critical to immunity and avoidance behavior (*Sym et al., 2000*; *Nagiel et al., 2008*; *Powell, 2008*). Notably, intestinal changes triggered by bacterial pathogen colonization activate the DAF-7/TGF-β pathway and the G-protein-coupled receptor (GPCR) NPR-1 pathway, which also regulates aerotaxis behavior (*Meisel and Kim, 2014*; *Styer et al., 2008*; *Singh and Aballay, 2019a*; *Singh and Aballay, 2019b*).

To uncover mechanisms that control immune and behavioral responses to invading pathogens independently of aerotaxis, we focused on studying mutants in *npr* genes that have not been previously linked to either host strategy against pathogen infection. Our investigation revealed that loss-of-function mutations in the NPR-15 encoding genes enhanced pathogen resistance when infected by Gram-negative and Gram-positive bacterial pathogens. Intriguingly, *npr-15* mutants exhibited a lack of pathogen avoidance behavior that was found to be independent of oxygen sensation. These findings point toward the involvement of a novel mechanism in the regulation of immune response and avoidance behavior.

Further analysis unveiled that the resistance to pathogen infection in *npr-15* mutants is mediated by the transcription factors GATA/ELT-2 and TFEB/HLH-30. These evolutionarily conserved transcription factors play vital roles in regulating immunity in *C. elegans* (*Head et al., 2017*; *Kerry et al., 2006*; *Olaitan and Aballay, 2018*; *Shapira et al., 2006*; *Visvikis et al., 2014*). Additionally, we discovered that NPR-15 controls avoidance behavior through the intestinal-expressed transient receptor potential melastatin (TRPM) ion channel, GON-2, which has recently been demonstrated to modulate avoidance behavior in Gram-positive bacteria (*Filipowicz et al., 2021*). Moreover, our results indicate that the amphid sensory neuron, ASJ, plays a crucial role in the interplay between immune response and avoidance behavior. These findings provide insights into the neural mechanisms that control immunity against bacterial infections and pathogen avoidance behavior.

## Results

### NPR-15 loss-of-function enhances pathogen resistance and inhibits avoidance behavior independently of aerotaxis

Out of 34 mutants in *npr* genes that were not previously linked to the control immunity (*Supplementary file 1A*), only animals lacking NPR-15 (*npr-15(tm12539)* and *npr-15(ok1626)* null animals) exhibited enhanced survival against *Pseudomonas aeruginosa*-mediated killing compared to wild-type (WT) animals (*Figure 1A*, *Figure 1—figure supplement 1A*, and *Supplementary file 1B*). Furthermore, we found that the *npr-15(tm12539)* animals exhibited less visible bacterial colonization and significantly reduced colony-forming units compared to WT animals (*Figure 1B and C*). The enhanced resistance to pathogen of *npr-15(ok1626)* animals appears to be universal, as the mutants were also found to be resistant to additional human pathogens, including Gram-negative *Salmonella enterica* strain 1344 and Gram-positive *Enterococcus faecalis* strain OG1RF and *Staphylococcus aureus* strain NCTCB325 (*Figure 1D–F*), suggesting that NPR-15 suppresses defense against bacterial pathogens in general. When exposed to live *Escherichia coli*, the primary food source of *C. elegans* in the laboratory, *npr-15(tm12539)* animals exhibited increased lifespan compared to WT animals (*Figure 1G*). However, there were no significant differences in longevity between *npr-15(tm12539)* and WT animals when they were exposed to lawns of *E. coli* that were rendered non-proliferating by ultraviolet light (UV) treatment (*Garigan et al., 2002*, *Figure 1H*). Expression of *npr-15* under the control of its own

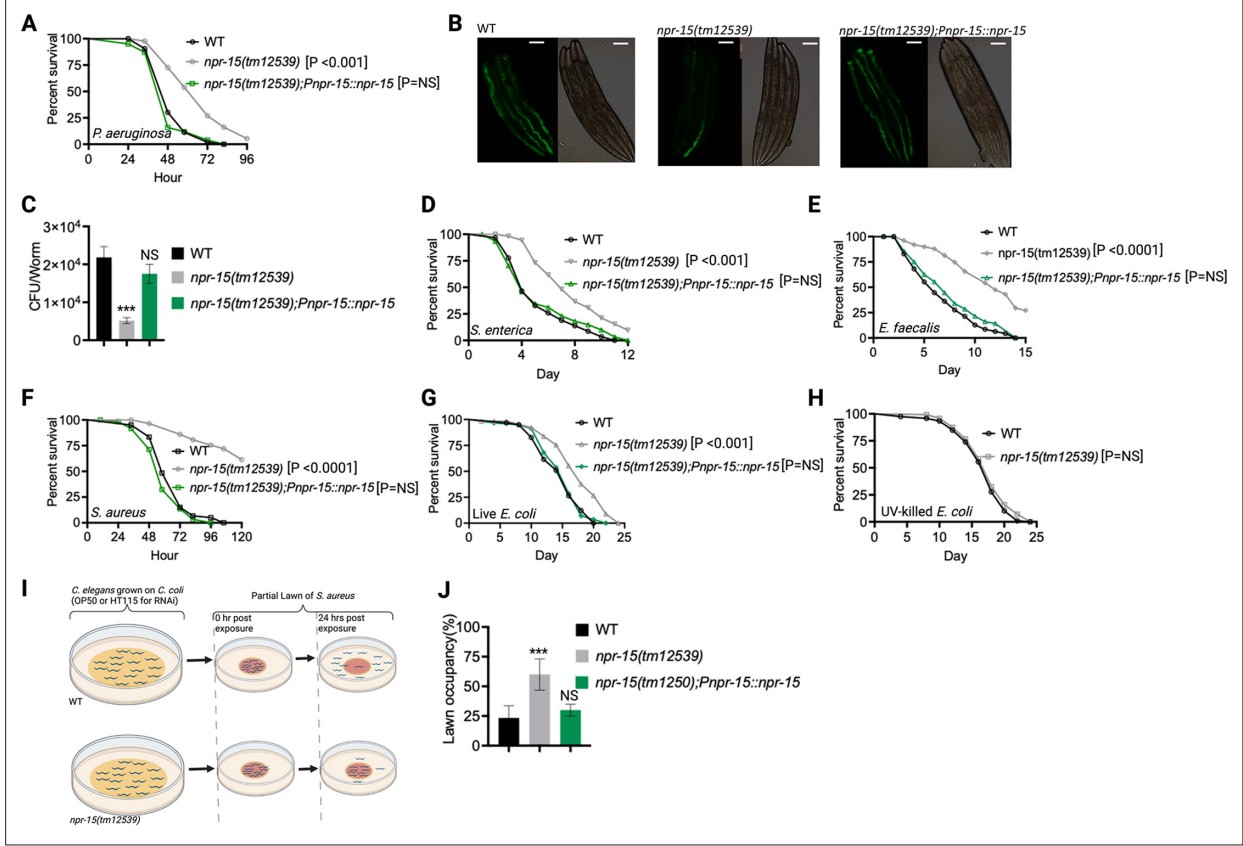

**Figure 1.** NPR-15 loss-of-function enhanced pathogen resistance and inhibited avoidance behavior. (**A**) Wild-type (WT), *npr-15(tm12539)*, and *npr-15(tm12539);Pnp-15::npr-15* animals were exposed to *P. aeruginosa* partial lawn and scored for survival. (**B**) Colonization of WT, *npr-15(tm12539)*, and *npr-15(tm12539);Pnp-15::npr-15* animals by *P. aeruginosa*-GFP after 24 hr at 25°C. Scale bar, 200 μm. (**C**) Colony-forming units per animal (WT, *npr-15(tm12539)*, *npr-15(tm12539);Pnp-15::npr-15*) grown on *P. aeruginosa*-GFP for 24 hr at 25°C. Bars represent means while error bars indicate the standard deviation (SD) of three independent experiments; ***$p<0.001$ and NS=not significant. (**D**) WT, *npr-15(tm12539)*, and *npr-15(tm12539);Pnp-15::npr-15* animals were exposed to *S. enterica* partial lawn and scored for survival. (**E**) WT, *npr-15(tm12539)*, and *npr-15(tm12539);Pnp-15::npr-15* animals were exposed to *E. faecalis* partial lawn and scored for survival. (**F**) WT, *npr-15(tm12539)*, and *npr-15(tm12539);Pnp-15::npr-15* animals were exposed to *S. aureus* partial lawn and scored for survival. (**G**) WT, *npr-15(tm12539)*, and *npr-15(tm12539);Pnp-15::npr-15* animals were exposed to live *E. coli* and scored for survival. (**H**) WT and *npr-15(tm12539)* animals were exposed to ultraviolet light (UV)-killed *E. coli* and scored for survival. (**I**) Schematics of avoidance behavior assay on *S. aureus*. (**J**) Lawn occupancy of WT *C. elegans* and *npr-15(tm12539)*, and *npr-15(tm1250);Pnpr-15::npr-15* animals on a partial lawn of *S. aureus* at 24 hr. Bars represent means while error bars indicate SD; **$p<0.001$ and NS=not significant.

The online version of this article includes the following figure supplement(s) for figure 1:

**Figure supplement 1.** NPR-15 loss-of-function exhibited pathogen resistance independent of brood size and oxygen-independent avoidance behavior.

promoter rescued the enhanced survival of *npr-15(tm12539)* animals to different bacterial pathogens (*Figure 1A–G*), indicating that the functional loss of NPR-15 enhanced the animal's survival against live bacteria.

Considering that *C. elegans* exhibits avoidance behavior when encountering certain pathogenic bacteria (*Meisel and Kim, 2014*; *Chang et al., 2011*; *Reddy et al., 2011*; *Singh and Aballay, 2019a*), we examined the lawn occupancy of *npr-15(tm12539)* and WT animals on the partial lawn of *S. aureus* cultured in the center of an agar plate (*Figure 1I*). Unexpectedly, we found that *npr-15(tm12539)* exhibited significantly reduced pathogen avoidance when exposed to *S. aureus* (*Figure 1J*). We also compared the re-occupancy of the lawn exhibited by WT and *npr-15(tm12539)* animals and found no differences in their re-occupancy (*Figure 1—figure supplement 1B*). Interestingly, we noticed that the variation in lawn occupancy is greater in WT than in *npr-15(tm12539)* animals across experiments (*Supplementary file 2*), which suggests that the strong lack of avoidance of *npr-15(tm12539)* somehow counteracts the experimental variation. We also found that *npr-15(tm12539)* exhibited reduced learned avoidance compared to WT animals (*Figure 1—figure supplement 1C*).

To investigate whether aerotaxis played a role in the lack of avoidance of *S. aureus* exhibited by *npr-15(tm12539)*, we studied lawn occupancy in the presence of 8% oxygen. As shown in *Figure 1—figure supplement 1D*, exposure to 8% oxygen to *npr-15(tm12539)* animals did not rescue the lack of avoidance to *S. aureus*, although it did rescue the lack of avoidance of *npr-1* mutants. Moreover, the survival of *npr-15(tm12539)* animals on full-lawn assays, where agar plates were completely covered by pathogenic bacteria to eliminate the possibility of pathogen avoidance, was significantly higher than that of WT animals (*Figure 1—figure supplement 1E and F*). These findings suggest that NPR-15 suppresses pathogen resistance and enhances avoidance behavior in response to pathogen infection, independently of oxygen concentrations.

Because *C. elegans* pharyngeal pumping directly affects bacterial intake (*Styer et al., 2008*; *Singh and Aballay, 2019a*; *Cao et al., 2017*; *Sellegounder et al., 2019*), we asked whether the resistance to infections and reduced pathogen avoidance behavior in *npr-15(tm12539)* animals could be attributed to a decrease in pathogen intake. We found that *npr-15(tm12539)* animals exhibited pumping rates comparable to that of WT animals (*Figure 1—figure supplement 1G*), indicating that the dose of pathogens is similar in both cases. Moreover, it has recently been demonstrated that bacterial accumulation in animals defective in the defecation motor program causes intestinal distension that elicits a robust immune response (*Singh and Aballay, 2019a*) and modulates pathogen avoidance (*Singh and Aballay, 2019a*; *Singh and Aballay, 2019b*; *Filipowicz et al., 2021*; *Kumar et al., 2019*; *Hong et al., 2021*). However, we found that the defecation cycle of *npr-15(tm12539)* animals is indistinguishable from that of WT animals (*Figure 1—figure supplement 1H*). Taken together, our results

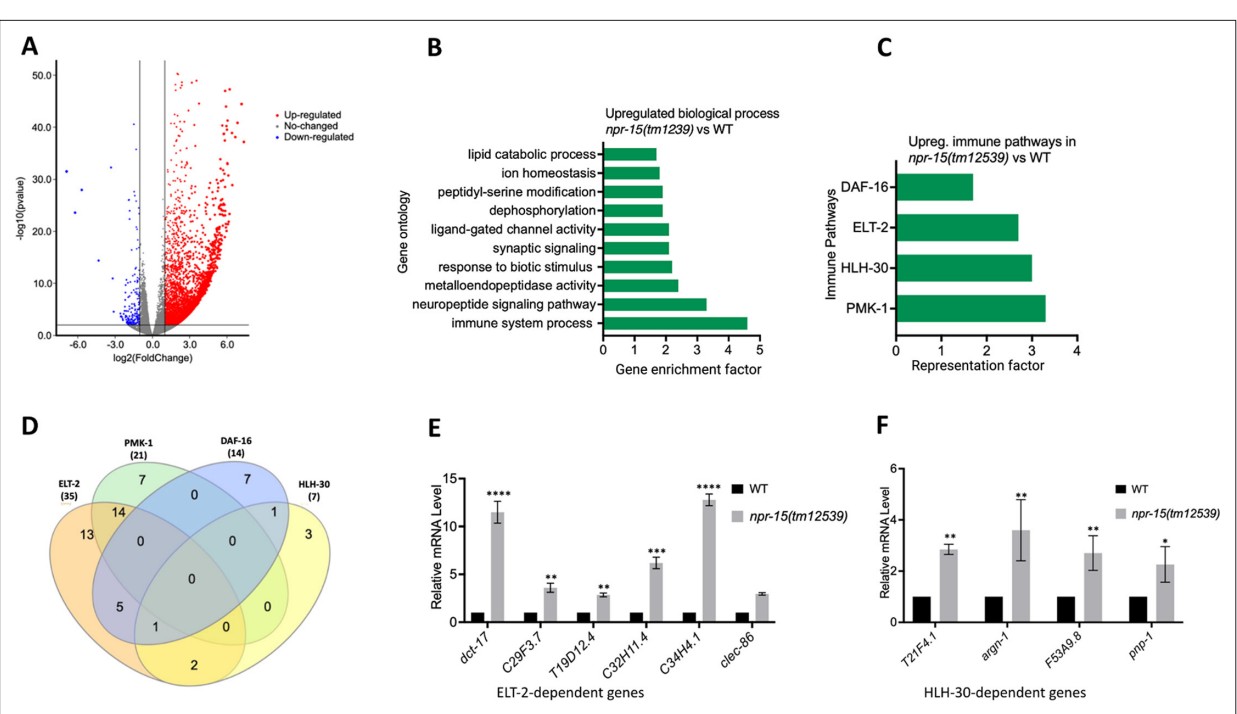

**Figure 2.** NPR-15 inhibits the expression of immune and aversion-related genes/pathways. (**A**) Volcano plot of upregulated and downregulated genes in *npr-15(tm12539)* vs. wild-type (WT) animals. Red and blue dots represent significant upregulated and downregulated genes respectively, while the gray dots represent not significant genes. (**B**) Gene ontology analysis of upregulated genes in *npr-15(tm12539)* vs. WT animals. The result was filtered based on significantly enriched terms, with a q value <0.1. (**C**) Representation factors of immune pathways for the upregulated immune genes in *npr-15(tm12539)* vs. WT animals. (**D**) Venn diagram showing the upregulated immune genes in each pathway in *npr-15(tm12539)* vs. WT animals. (**E**) Quantitative reverse transcription-PCR (qRT-PCR) analysis of ELT-2-depenent immune gene expression in WT and *npr-15(tm12539)* animals. Bars represent means while error bars indicate standard deviation (SD) of three independent experiments; *p<0.05, **p<0.001, and ***p<0.0001. (**F**) qRT-PCR analysis of HLH-30-depenent immune gene expression in WT and *npr-15(tm12539)* animals. Bars represent means while error bars indicate SD of three independent experiments; *p<0.05, **p<0.001, and ***p<0.0001.

The online version of this article includes the following figure supplement(s) for figure 2:

**Figure supplement 1.** Downregulated biological process and upregulated immune pathways/gene number in *npr-15(tm12539)* animals.

show that NPR-15 loss-of-function enhances pathogen resistance and inhibits avoidance behavior, suggesting NPR-15 suppresses molecular immunity while activating behavioral immunity.

## The loss of NPR-15 leads to the upregulated immune and neuropeptide genes

To understand the immune mechanisms controlled by NPR-15 in defense against pathogen exposure, we conducted transcriptomic analyses to identify dysregulated genes in *npr-15(tm12539)* compared to WT animals (*Figure 2A* and *Supplementary file 3*). To identify gene groups that were controlled by NPR-15, we performed an unbiased gene enrichment analysis using a WormBase enrichment analysis tool (https://wormbase.org/tools/enrichment/tea/tea.cgi) that is specific for *C. elegans* gene data analyses (*Angeles-Albores et al., 2016*). The study revealed 10 ontology clusters with high enrichment scores of vital biological functions for upregulated and downregulated genes in *npr-15(tm12539)* (*Figure 2B*, *Figure 2—figure supplement 1*). Overall, the gene expression data showed significant upregulation of immune/defense response and neuropeptide signaling pathway genes (*Figure 2B*). Genes associated with synaptic signaling, ligand-gated channel activity, lipid metabolism, and response to biotic stimuli were also upregulated in *npr-15(tm12539)* animals.

To further explore potential immune pathways involved in pathogen resistance in NPR-15-deficient animals, we performed a gene enrichment analysis using the immune gene subset with the Worm Exp tool (https://wormexp.zoologie.uni-kiel.de/wormexp/) (*Yang et al., 2016*). This tool integrates all published expression datasets for *C. elegans* to analyze and generate pathways. Our analysis revealed a high enrichment of pathways crucial for *C. elegans* defense against bacterial infections, including the ELT-2, HLH-30, PMK-1, and DAF-2/DAF-16 insulin pathways (*Figure 2C* and *Supplementary file 4*, *Kerry et al., 2006*; *Shapira et al., 2006*; *Visvikis et al., 2014*; *Kim et al., 2002*; *Aballay et al., 2003*; *Garsin et al., 2003*; *Hsu et al., 2003*; *Murphy et al., 2003*; *Huffman et al., 2004*). Notably,

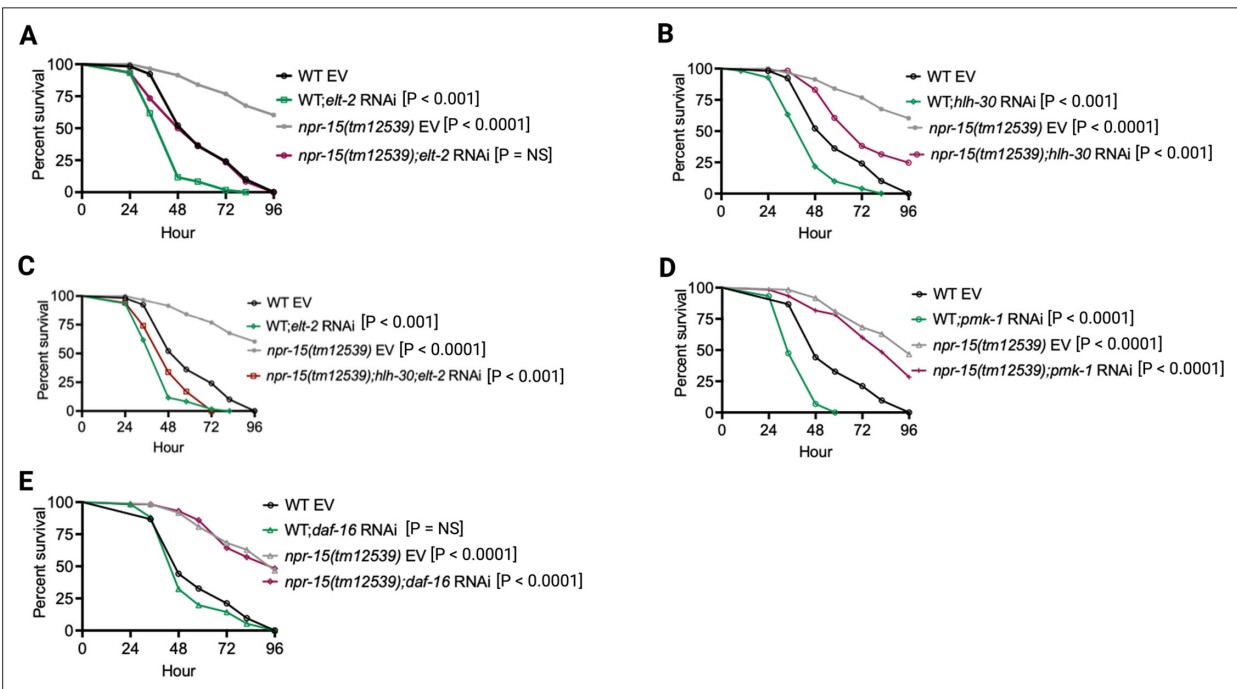

**Figure 3.** NPR-15 loss-of-function enhances immunity via ELT-2 and HLH-30 when exposed to *S. aureus*. (**A**) Wild-type (WT) and *npr-15(tm12539)* animals fed with *elt-2* RNAi were exposed to *S. aureus* full lawn and scored for survival. EV, empty vector RNAi control. (**B**) WT and *npr-15(tm12539)* animals fed with *hlh-30* RNAi were exposed to *S. aureus* full lawn and scored for survival. EV, empty vector RNAi control. (**C**) WT and *npr-15(tm12539)* animals fed with *hlh-30* and *elt-2* RNAi were exposed to *S. aureus* full lawn and scored for survival. EV, empty vector RNAi control. (**D**) WT and *npr-15(tm12539)* animals fed with *pmk-1* RNAi and animals were exposed to *S. aureus* full lawn and scored for survival. EV, empty vector RNAi control. (**E**) WT and *npr-15(tm12539)* animals fed with *daf-16* RNAi were exposed to *S. aureus* full lawn and scored for survival. EV, empty vector RNAi control.

The online version of this article includes the following figure supplement(s) for figure 3:

**Figure supplement 1.** NPR-15 loss-of-function enhances immunity via ELT-2 and HLH-30 when exposed to *S. aureus*.

we observed that several of the PMK-1-, HLH-30-, and DAF-16-dependent genes were also controlled by ELT-2 (*Figure 2D*), suggesting a potentially major role of ELT-2-dependent genes in the enhanced resistance to the pathogen phenotype of *npr-15(tm12539)* animals. To validate these findings, we performed quantitative PCR and confirmed the upregulation of ELT-2-dependent immune genes such as *dct-17*, C29F3.7, T19D12.4, C32H11.4, C34H4.1, and *clec-86* (*Figure 2E*). We also validated HLH-30-dependent genes T21F4.1, *argn-1*, F53A9.8, and *pnp-1* (*Figure 2F*). These transcriptomic and gene analyses indicate that NPR-15 primarily regulates *C. elegans* defense against bacterial infections by inhibiting immune genes, several of which are controlled by ELT-2, HLH-30, and PMK-1.

## NPR-15 controls ELT-2- and HLH-30-dependent genes via sensory neuron, ASJ

To investigate whether the enhanced resistance to pathogen infection of *npr-15(tm12539)* animals is due to the upregulation of immune genes, we examined the role of RNA interference (RNAi)-mediated suppression of the immune pathways shown in *Figure 2C*. We inactivated *elt-2*, *pmk-1*, *hlh-30*, and *daf-16* in WT and *npr-15(tm12539)* animals and exposed them to *S. aureus*. We found that *elt-2* RNAi completely suppressed the enhanced resistance to *S. aureus* infection in *npr-15(tm12539)* animals (*Figure 3A*). Partial suppression of pathogen resistance in *npr-15(tm12539)* animals was observed with *hlh-30* RNAi (*Figure 3B*). Furthermore, when both *hlh-30* and *elt-2* were inactivated in *npr-15(tm12539)* animals and were exposed to *S. aureus*, their susceptibility to infection was comparable to that of *elt-2* RNAi animals (*Figure 3C*). However, *pmk-1*, *daf-16* RNAi failed to suppress the pathogen resistance of *npr-15(tm12539)* (*Figure 3D and E*). To confirm these findings, we crossed mutants in the aforementioned immune regulators to *npr-15(tm12539)* animals and exposed them to *S. aureus* (*Figure 3—figure supplement 1A–C*). We also studied the aforementioned immune

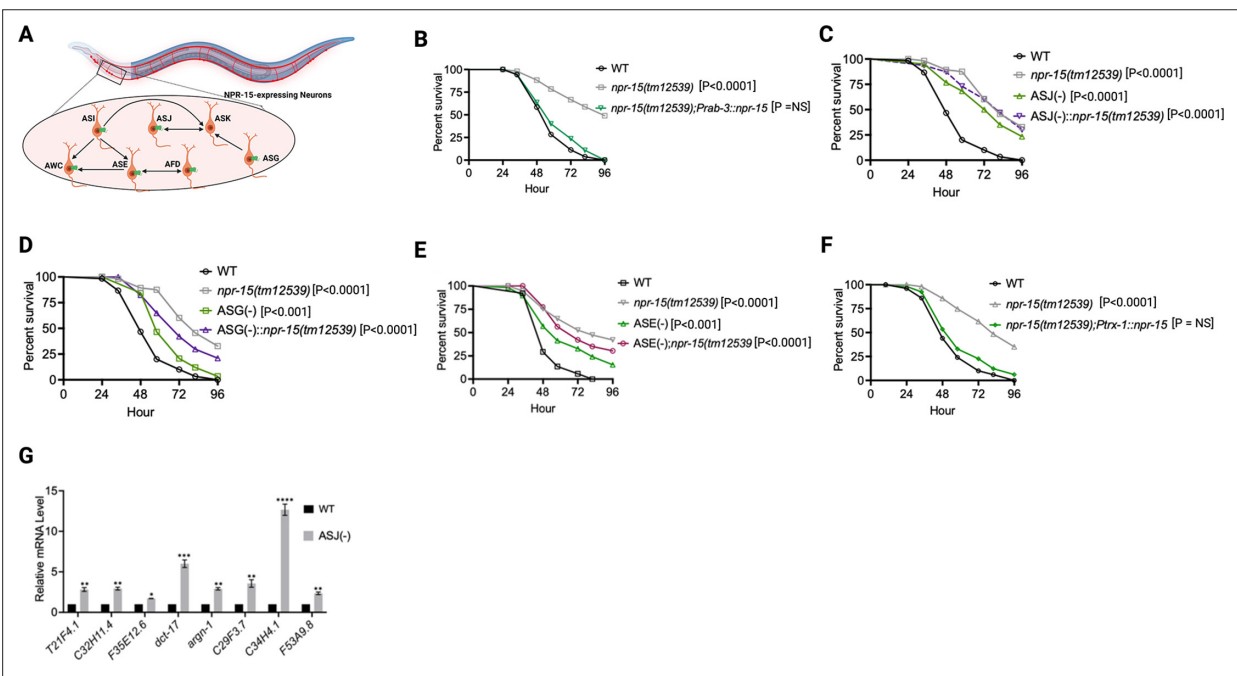

**Figure 4.** NPR-15 suppresses molecular immune response via sensory neurons, ASJ. (**A**) Neuronal connectome of NPR-15-expressing neurons. (**B**) Wild-type (WT), *npr-15(tm12539)*, and *npr-15(tm12539);Prab-3::npr-15* animals were exposed to *S. aureus* full lawn and scored for survival. (**C**) WT, *npr-15(tm12539)*, ASJ(-), and ASJ(-);*npr-15(tm12539)* animals were exposed to *S. aureus* full lawn and scored for survival. (**D**) WT, *npr-15(tm12539)*, ASG(-), and ASG(-);*npr-15(tm12539)* animals were exposed to *S. aureus* full lawn and scored for survival. (**E**) WT, *npr-15(tm12539)*, ASE(-), and ASE(-);*npr-15(tm12539)* animals were exposed to *S. aureus* full lawn and scored for survival. (**F**) WT, *npr-15(tm12539)*, *npr-15(tm12539);Ptrx-1::npr-15* animals were exposed to *S. aureus* full lawn and scored for survival. (**G**) Quantitative reverse transcription-PCR (qRT-PCR) analysis of ELT-2- and HLH-30-depenent immune gene expression in WT and ASJ(-) animals. Bars represent means while error bars indicate standard deviation (SD) of three independent experiments; *p<0.05, **p<0.001, and ***p<0.0001.

The online version of this article includes the following figure supplement(s) for figure 4:

**Figure supplement 1.** Survival of other NPR-15-expressing neuron-ablated strains in *S. aureus* infection.

pathways in the response of *npr-15(tm12539)* animals to *P. aeruginosa* (**Figure 3—figure supplement 1D–F**). We found that *hlh-30* or *pmk-1* mutation partly suppressed the resistance to *P. aeruginosa* infection phenotype of *npr-15(tm12539)* animals (**Figure 3—figure supplement 1D and E**). These results indicate that NPR-15 suppresses ELT-2- and HLH-30-dependent molecular immunity.

Because NPR-15 is expressed in six sensory neuronal cells (ASG, ASI, ASJ, ASE, AFD, and AWC) as well as an interneuron, AVK (**Figure 4A**, **Harris et al., 2020**), we studied whether the NPR-15-expressing cells could control the defense response against pathogen infection. First, we examined the neuronal connectome and communication between the NPR-15-expressing cell and found well-established synaptic connections between all the sensory neurons (**Figure 4A**). Next, we asked if NPR-15 could be acting in a neuron-intrinsic manner to control immune response. To address this question, we specifically inactivated *npr*-15 in neurons and the intestine using tissue-specific RNAi strains and assessed the survival of the animals after *S. aureus* infection. Unlike intestine-specific RNAi (strain MGH171) (**Figure 4—figure supplement 1A**), neural-specific RNAi (strain TU3401) of NPR-15 resulted in a pathogen resistance phenotype similar to that of *npr-15(tm12539)* animals (**Figure 4—figure supplement 1B**). This finding was further supported by rescuing NPR-15 under the control of a pan-neuronal promoter and exposing the animals to *S. aureus*. The results demonstrated that pan-neuronal promoter-driven expression of NPR-15 rescued the enhanced survival phenotype of *npr-15(tm12539)* animals (**Figure 4B**). These results suggest that NPR-15 suppresses immunity through the nervous system. We next studied the specific NPR-15-expressing neuronal cells (ASG, ASI, ASJ, ASE, AFD, AWC) that could control the defense response against pathogen infection. To identify the neuronal cells responsible for the NPR-15-mediated immune control, we crossed strains lacking the neurons with *npr-15(tm12539)* animals and studied their effect on defense against pathogen infection (**Figure 4C–E** and **Figure 4—figure supplement 1C–E**). We found that ASJ(-) animals exhibited resistance to pathogen-mediated killing similar to that of *npr-15(tm12539)* animals (**Figure 4C**). Although the ASG(-) and ASE(-) neuron-ablated strains demonstrated pathogen resistance phenotype, there is a significant difference compared to that of *npr-15(tm12539)* animals (**Figure 4D–E**). Hence, we further confirmed the role of NPR-15/ASJ neurons in suppressing immunity by rescuing NPR-15 on ASJ unique promoter and performing survival experiments with the rescued strain. Results showed that the ASJ-specific rescue of NPR-15 successfully blocked the enhanced survival of *npr-15(tm12539)* animals (**Figure 4F**). We also quantified the expression of immune genes and found that they were upregulated in ASJ(-) animals (**Figure 4G**). Collectively, these findings suggest that NPR-15 controls immunity in a manner similar to that of ASJ neurons.

## The lack of avoidance behavior by NPR-15 loss-of-function is independent of immunity and neuropeptide genes

Having established the upregulation of immune genes in *npr-15(tm12539)* animals compared to WT animals (**Figure 2B–F** and **Supplementary file 4**), we determine whether the reduced pathogen avoidance of *npr-15(tm12539)* animals could be attributed to the upregulation of immune pathways. To investigate this, we employed RNAi to suppress immune transcription factors/regulators and evaluated their impact on pathogen avoidance behavior in both WT and *npr-15(tm12539)* animals. Our results indicate that none of the tested immune regulators (*elt-2, pmk-1, daf-16,* and *hlh-30*) were able to suppress the lack of pathogen avoidance behavior observed in response to *S. aureus* (**Figure 5—figure supplement 1A–D**). Furthermore, we inactivated immune genes that are not controlled by the immune regulators tested above, but none of them were able to suppress the lack of avoidance behavior in the *npr-15(tm12539)* animals (**Supplementary file 5**). Given the possibility of functional redundancy among these genes, we cannot rule out the possibility that different combinations may play a role in controlling avoidance behavior. These findings indicate that the avoidance behavior observed in *npr-15(tm12539)* is independent of individual immune genes upregulated in *npr-15(tm12539)* animals.

Additionally, previous studies have shown that neuropeptides expressed in the intestine can modulate avoidance behavior (**Lee and Mylonakis, 2017**), and we found that neuropeptides are among the most highly upregulated genes in *npr-15(tm12539)* animals (**Figure 2B** and **Supplementary file 6**). To study whether an intestinal signal may act through NPR-15 to regulate avoidance, we inactivated the upregulated intestinal neuropeptide genes in *npr-15(tm12539)* animals. Our experiments revealed that none of the inactivated intestinal-expressed neuropeptides were able to suppress the

lack of avoidance behavior of *npr-15(tm12539)* animals in response to *S. aureus* (*Figure 5—figure supplement 1E* and *Supplementary file 5*). Therefore, it can be concluded that the absence of avoidance behavior by loss of NPR-15 function is independent of both immune and intestinal neuropeptide signaling pathways.

## NPR-15 controls pathogen avoidance via sensory neuron, ASJ, and intestinal TRPM channel, GON-2

Having demonstrated that NPR-15 controls immune response via sensory neurons, ASJ (*Figure 4C and F–G*), we sought to identify the neurons involved in the lack of avoidance behavior to *S. aureus* observed in *npr-15(tm12539)* animals. First, we investigated the avoidance behavior of a pan-neuronal rescued strain of NPR-15, which successfully rescued the suppressed pathogen avoidance of *npr-15(tm12539)* animal (*Figure 5A*). Next, we asked which of the NPR-15-expressing-neuronal cells could control pathogen avoidance. To answer this question, we evaluated the pathogen avoidance of the different neuron-ablated strains that were crossed with *npr-15(tm12539)* animals. Consistent with our previous findings, we found that only ASJ(-) exhibited reduced pathogen avoidance similar to that of *npr-15(tm12539)* animals (*Figure 5B*). This observation was further confirmed by rescuing NPR-15 under the control of an ASJ unique promoter (*Figure 5C*). The pathogen avoidance of other neuronal-ablated strains was comparable to that of WT animals (*Figure 5D–G*). These results suggest that the loss of NPR-15 function suppresses behavioral immunity via sensory neurons, specifically ASJ.

Because TRPM ion channels, GON-2 and GTL-2, are required for pathogen avoidance (*Filipowicz et al., 2021*), we studied whether they may be part of the NPR-15 pathway that controls pathogen avoidance. We inactivated *gon-2* and *gtl-2* in *npr-15(tm12539)* animals and WT animals. Our findings showed that only *gon-2* null animals, but not *gtl-2*, exhibited pathogen avoidance behavior similar to that of *npr-15(tm12539)* animals (*Figure 5H–I*). This suggests that both NPR-15 and GON-2 may function in a shared pathway to regulated pathogen avoidance behavior. As it has been previously demonstrated that GON-2 modulates avoidance behavior through the intestine (*Filipowicz et al., 2021*), we confirmed the role of intestinal-expressed GON-2 in pathogen avoidance by inactivating *gon-2* in an RNAi intestine-specific strain (MGH171), as well as in an RNAi neuron-specific strain (TU3401) that was used as a control. These animals were exposed to *S. aureus* to study their lawn occupancy. Our results showed that the inactivation of *gon-2* in MGH171 and MGH171;*npr-15(tm12539)* (*Figure 5J*), but not in TU3401 (*Figure 5—figure supplement 2*), exhibited comparable avoidance behaviors to that of *npr-15(tm12539)* animals. These results suggest that NPR-15 acts through the intestinal TRPM channel GON-2 to control pathogen avoidance behavior. Since we have previously shown that only ASJ(-) animals among other neuron-ablated strains exhibit a similar avoidance behavior to *npr-15(tm12539)* animals (*Figure 5B and D–G*), we investigated whether NPR-15 control of avoidance behavior toward *S. aureus* in a GON-2-dependent manner involves ASJ. We used RNAi to inactivate *gon-2* in ASJ(-), ASJ(-);*npr-15(tm152799)*, *npr-15(tm12539)*, and WT animals before exposing them to *S. aureus* to assay their lawn occupancy. Our results showed that the avoidance behavior of ASJ(-);*gon-2* is comparable to that of ASJ(-), ASJ(-);*npr-15(tm12539)*, and *gon-2* null animals when exposed to *S. aureus* (*Figure 5K*). These results suggest that NPR-15 controls *S. aureus* avoidance in a GON-2-dependent manner via the ASJ neuron. In summary, the neuronal GPCR/NPR-15 plays a dual role: suppressing the immune response and enhancing avoidance behavior toward *S. aureus* through sensory neurons, specifically ASJ. The control of immunity against the pathogen *S. aureus* is dependent on the ELT-2 and HLH-30 transcription factors, while the control of avoidance behavior is GON-2-dependent and mediated through the intestine (*Figure 6*).

## Discussion

GPCRs play a crucial role in neuronal and non-neuronal tissues in shaping both immune and avoidance behavioral responses toward invading pathogens (*Furness and Sexton, 2017*). In this study, we investigated the dual functionality of GPCR/NPR-15 in regulating molecular innate immunity and pathogen avoidance behavior independently of aerotaxis. Additionally, we aimed to understand the mechanism by which the nervous system controls the interplay between these crucial survival strategies in response to pathogenic threats. Our findings indicate that NPR-15 suppresses immune responses against different pathogen infections by inhibiting the activity of the GATA/ELT-2 and HLH-30 immune

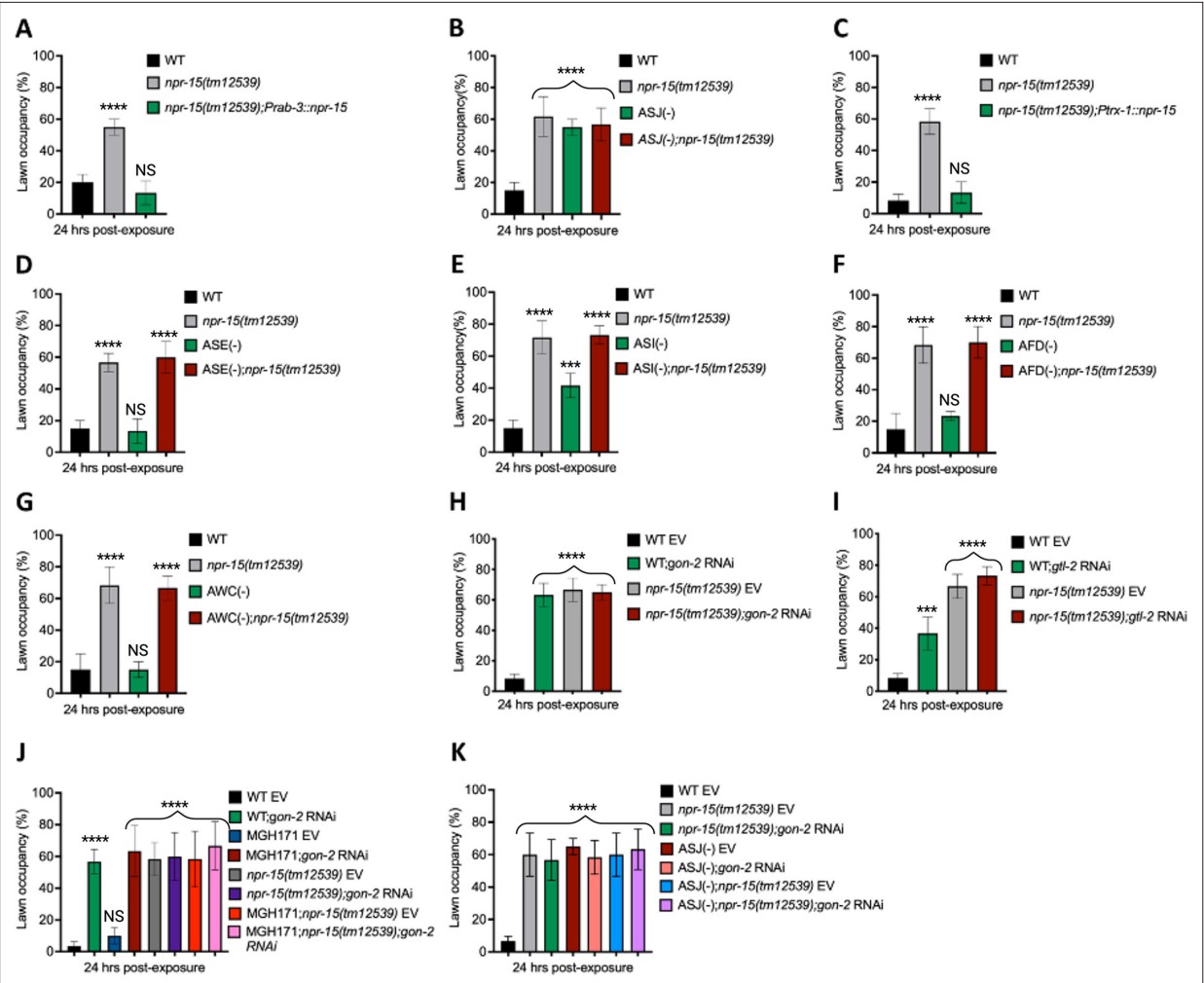

**Figure 5.** NPR-15 loss-of-function inhibits pathogen avoidance behavior in an ASJ-dependent manner. (**A**) Wild-type (WT), *npr-15(tm12539),* and *npr-15(tm12539);Prab-3::npr-15* animals were exposed to partial lawn of *S. aureus* and scored for lawn occupancy. Bars represent means while error bars indicate the standard deviation (SD) of three independent experiments; ***p<0.0001 and NS=not significant. (**B**) WT, *npr-15(tm12539),* ASJ(-), and ASJ(-);*npr-15(tm12539)* animals were exposed to partial lawn of *S. aureus* and scored for lawn occupancy. Bars represent means while error bars indicate the SD of three independent experiments; ***p<0.0001 and NS=not significant. (**C**) WT, *npr-15(tm12539),* and *npr-15(tm12539);Ptrx-1::npr-15* animals were exposed to partial lawn of *S. aureus* and scored for lawn occupancy. Bars represent means while error bars indicate the SD of three independent experiments; ***p<0.0001 and NS=not significant. (**D**) WT, *npr-15(tm12539),* ASE(-), and ASE(-);*npr-15(tm12539)* animals were exposed to partial lawn of *S. aureus* and scored for lawn occupancy. Bars represent means while error bars indicate the SD of three independent experiments; ***p<0.0001 and NS=not significant. (**E**) WT, *npr-15(tm12539),* ASI(-), and ASI(-);*npr-15(tm12539)* animals were exposed to partial lawn of *S. aureus* and scored for lawn occupancy. Bars represent means while error bars indicate the SD of three independent experiments; **p<0.001, ***p<0.0001, and NS=not significant. (**F**) WT, *npr-15(tm12539),* and AFD(-), and AFD(-);*npr-15(tm12539)* animals were exposed to partial lawn of *S. aureus* and scored for lawn occupancy. Bars represent means while error bars indicate the SD of three independent experiments; ***p<0.0001 and NS=not significant. (**G**) WT, *npr-15(tm12539),* and AWC(-), and AWC(-);*npr-14(tm12539)* animals were exposed to partial lawn of *S. aureus* and scored for lawn occupancy. Bars represent means while error bars indicate the SD of three independent experiments; ***p<0.0001 and NS=not significant. (**H**) WT and *npr-15(tm12539)* fed with *gon-2* RNAi and were exposed to the partial lawn of *S. aureus* and scored for lawn occupancy. EV, empty vector RNAi control. Bars represent means while error bars indicate the SD of three independent experiments; ***p<0.0001. (**I**) WT and *npr-15(tm12539)* fed with *gtl-2* RNAi and were exposed to the partial lawn of *S. aureus* and scored for lawn occupancy. EV, empty vector RNAi control. Bars represent means while error bars indicate the SD of three independent experiments; **p<0.001, ***p<0.0001, and NS=not significant. (**J**) WT, *npr-15(tm12539),* RNAi intestine-specific strain MGH171, MGH171; *npr-15(tm12539)* fed with *gon-2* RNAi and were exposed to the partial lawn of *S. aureus* and scored for lawn occupancy. EV, empty vector RNAi control. Bars represent means while error bars indicate the SD of three independent experiments; ***p<0.0001 and NS=not significant. (**K**) WT, *npr-15(tm12539),* ASJ(-), ASJ(-);*npr-15(tm12539)* fed with *gon-2* RNAi and were exposed to the partial lawn of *S. aureus* and scored for lawn occupancy. EV, empty vector RNAi control. Bars represent means while error bars indicate the SD of three independent experiments; ***p<0.0001.

The online version of this article includes the following figure supplement(s) for figure 5:

*Figure 5 continued on next page*

*Figure 5 continued*

**Figure supplement 1.** NPR-15 controls avoidance behavior independent of immunity, neuropeptide pathways, and oxygen.

**Figure supplement 2.** The transient receptor potential melastatin (TRPM) channel GON-2 control avoidance is independent of the nervous system.

transcription regulators. Moreover, we found that the control of the pathogen avoidance behavior by NPR-15 is independent of aerotaxis and dependent on intestinal GON-2. Furthermore, we demonstrated that NPR-15 controls the immunity-behavioral response via an amphid sensory neuron, ASJ, in response to *S. aureus* infection.

ASJ neurons are known to regulate different biological functions such as lifespan (**Alcedo and Kenyon, 2004**), dauer activities (**Bargmann and Horvitz, 1991**; **Chung et al., 2013**), head-directed light avoidance (**Ward et al., 2008**), and food search (**Macosko et al., 2009**). We show here that ASJ neurons play a role in the control of immune and behavioral responses against pathogen infection through GPCR/NPR-15. Our research additionally indicates that the regulation of NPR-15-mediated avoidance is not influenced by intestinal immune and neuropeptide genes. Given the potential for functional redundancy and our focus on genes upregulated in the absence of NPR-15, we cannot entirely rule out the possibility that unexamined immune effectors or neuropeptides, not transcriptionally controlled by NPR-15, might be involved. Different intestinal signals may also participate in the NPR-15 pathway that controls pathogen avoidance.

Other neural GPCRs have also been linked to the control of innate immunity in *C. elegans* (**Styer et al., 2008**; **Frooninckx et al., 2012**; **Gupta and Singh, 2017**). For example, the octopamine receptor OCTR-1 (**Sun et al., 2011**), the olfactory learning receptor OLRN-1 (**Foster et al., 2020**), and the D1-like dopamine receptor DOP-4 (**Cao and Aballay, 2016**) function in ASH/ASI, AWC, and ASG neurons, respectively. GPCRs that are neuropeptide receptors, such as NPR-1 (**Styer et al., 2008**),

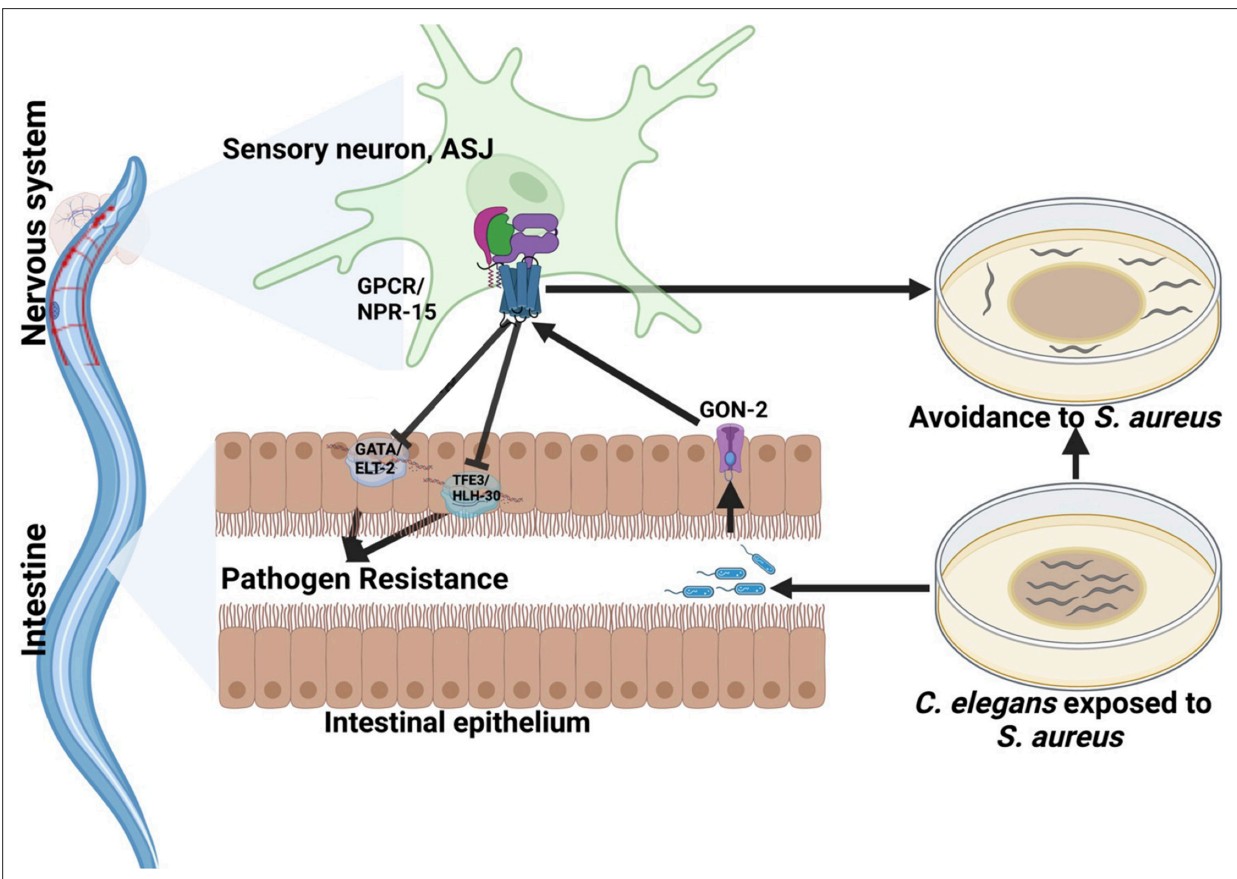

**Figure 6.** G-protein-coupled receptor (GPCR)/NPR-15 suppressed immune response and enhanced avoidance behavior via sensory neurons, ASJ. The immune response control is dependent on ELT-2 and HLH-30 transcription factors, while NPR-15 controls avoidance behavior to *S. aureus* via intestinal-expressed transient receptor potential melastatin (TRPM) channel, GON-2.

NPR-9 (*Yu et al., 2018*), and NPR-8 (*Sellegounder et al., 2019*), also function in neurons to control immunity. In common, these GPCRs regulate innate immunity via the p38/PMK-1 MAPK pathway (*Styer et al., 2008*; *Cao et al., 2017*; *Sellegounder et al., 2019*; *Foster et al., 2020*; *Cao and Aballay, 2016*; *Yu et al., 2018*). In contrast, NPR-15 functions in ASJ neurons to suppress immune defense against bacterial infections by inhibiting the transcriptional activity of ELT-2 and HLH-30. Two of these GPCRs, NPR-1 (*Styer et al., 2008*) and PCDR-1 (*Anderson et al., 2019*), also play a role in mediating pathogen avoidance. While OCTR-1-expressing neurons ASI play a role in avoidance (*Cao et al., 2017*), the specific role of OCTR-1 in ASH and ASI neurons remains unclear.

Our findings shed light on the role of NPR-15 in the control of the immune response. NPR-15 seems to suppress specific immune genes while activating pathogen avoidance behavior to minimize potential tissue damage and the metabolic energy cost associated with activating the molecular immune response against pathogen infections. Overall, the control of immune activation is essential for maintaining homeostasis and preventing excessive tissue damage caused by an overly aggressive and energy-costly response against pathogens (*Martin et al., 2017*; *Otarigho and Aballay, 2021*; *Ganeshan and Chawla, 2014*; *Ganeshan et al., 2019*).

## Conclusion

Our research uncovers the dual regulatory role of NPR-15 in both immunity and avoidance behavior, independent of aerotaxis, mediated by amphid sensory neurons. The host relies on behavioral responses to minimize or completely avoid pathogen exposure, effectively preventing the activation of the immune pathways and production of immune effector molecules, which can be metabolically costly. Moreover, the sustained and prolonged activation of the molecular immune system can have detrimental effects on the host. Understanding the organismal control of molecular and behavioral immune responses to pathogens can provide valuable insights into universal mechanisms used across species to maintain homeostasis during infections.

# Materials and methods
## Bacterial strains

The bacterial strains used in this study are *E. coli* OP50, *E. coli* HT115(DE3) (*Brenner, 1974*), *P. aeruginosa* PA14, *P. aeruginosa* PA14-GFP (*Tan et al., 1999a*; *Tan et al., 1999b*), *S. enterica* serovar Typhimurium 1344 (*Wray and Sojka, 1978*), and *S. aureus* strain NCTCB325 (*Sifri et al., 2003*). Gram-negative bacteria were grown in Luria-Bertani (LB) broth. *S. aureus* strain NCTCB325 was grown in Tryptic Soy Agar prepared with nalidixic acid. All bacteria were grown at 37°C.

### *C. elegans* strains and growth conditions

Hermaphrodite *C. elegans* (var. Bristol) WT was used as control unless otherwise indicated. *C. elegans* strains RB1429 *npr-15(ok1626)*, CX14394 *npr-5(ok1583)*, RB1836 *npr-14(ok2375)*, RB1289 *npr-18(ok1388)*, RB1405 *npr-22(ok1598)*, VC2526 *npr-35(ok3258)*, DA609 *npr-1(ad609)*, IM222 *npr-1(ur89)*, CX4148 *npr-1(ky13)*, CZ9957 *gtl-2(n2618)*, EJ1158 *gon-2(q388)*, EJ26 *gon-2(q362)*, LH202 *gtl-2(tm1463)*, JIN1375 *hlh-30(tm1978)*, CF1038 *daf-16(mu86)*, KU25 *pmk-1(km25)*, RB1668 *C02H7.2(ok2068)*, RB1632 *T02E9.1(ok2008)*, VC2421 *R106.2(ok3192)*, PS8315 *npr-29(sy1270)*, RB1365 *npr-16(ok1541)*, RB1958 *T07D4.1(ok2575)*, PS8484 *npr-31(sy1360)*, RB1429 *npr-15(ok1626)*, RB1325 *C53C7.1(ok1442)*, RB1938 *Y116A8B.5(ok2541)*, PS8177 *npr-23(sy1203)*, PS8317 *npr-33(sy1272)*, PS8450 *npr-27(sy1315)*, PS8444 *npr-21(sy1309)*, PS8442 *npr-26(sy1307)*, CX14394 *npr-5(ok1583)*, RB1836 *W05B5.2(ok2375)*, RB1405 *Y59H11AL.1(ok1598)*, MGH171 *alxIs9* [*vha-6p::sid-1::SL2::GFP*], TU3401 *uIs69* [*pCFJ90* (*myo-2p::mCherry*)+*unc-119p::sid-1*], DA609 *npr-1(ad609)*, DA508 *npr-1(n1353)*, GR2245 *skn-1(mg570)*, JPS582 *vxEx582* [*gcy-8p::ICE+myo-2p::mCherry*] [AFD(-)], PR680 *che-1(p680)* [ASE(-)], PR674 *che-1(p674)* [ASE(-)], PR672 *che-1(p672)* [ASE(-)], JN1713 *peIs1713* [*sra-6p::mCasp-1+unc-122p::mCherry*] [ASH(-)], KJ412 *trx-1(jh127)* [ASJ(-)], VZ1 *trx-1(ok1449)* [ASJ(-)], RB1332 *trx-1(ok1449)* [ASJ(-)], JN1715 *peIs1715* [*str-1p::mCasp-1+unc-122p::GFP*] [AWB(-)], PS6025 *qrIs2* [*sra-9::mCasp1*] [ASK(-)], and PY7505 *oyIs84* [*gpa-4p::TU#813+gcy-27p::TU#814+gcy-27p::GFP+unc-122p::DsRed*] [ASI(-)] were obtained from the Caenorhabditis Genetics Center (University of Minnesota, Minneapolis, MN, USA). We also obtained FX22538 *npr-15(tm12539)*, FX21950 *npr-3(tm1583)*, FX01583 *npr-3(tm11950)*, FX01782 *npr-4(tm1782)*, FX01497 *npr-6(tm1497)*, FX01498

*npr-12(tm1498)* and FX01504 *npr-13(tm1504)*, FX01665 *npr-34(tm1665)*, FX31809 *npr-30(tm6617)*, FX21079 *npr-28(tm1155)*, FX22538 *npr-15(tm12539)*, FX21950 *npr-3(tm11950)*, FX01583 *npr-3(tm1583)*, FX01782 *npr-4(tm1782)*, FX01497 *npr-6(tm1497)*, FX01498 *npr-12(tm1498)*, FX01504 *npr-13(tm1504)*, FX03170 *npr-17(tm3170)*, FX03225 *npr-17(tm3225)*, FX03170 *npr-17(tm3170)*, FX03225 *npr-17(tm3225)*, FX31312 *npr-22(tm8953)* from National Bioresource Project (NBRP), Japan. ASE(-);*npr-15(tm12539)*, ASI(-);*npr-15(tm12539)*, ASJ(-);*npr-15(tm12539)*, ASG(-);*npr-15(tm12539)*, *daf-16(mu86)*; *npr-15(tm12539)*, *pmk-1(km25)*;*npr-15(tm12539)*, *hlh-30(tm1978)*;*npr-15(tm12539)*, *skn-1(mg570)*;*npr-15(tm12539)*, and ASG(-):*npr-15(tm12539)* were obtained by standard genetic crosses. Rescued strain *npr-15(tm12539)*;*Pnpr-15::npr-15*, neuronal rescued strain *npr-15(tm12539)*;*Prab-3::npr-15* and *npr-15* rescued in ASJ neuron-specific, *npr-15(tm12539)*;*Ptrx-1::npr-15* were generated as described below. The ASG(-)-ablated Ex[*Ptax-2::CZ::ced-3(p17)::unc-54* 3'UTR+*Plim-6::ced-3(p15)-NZ::unc-54* 3'UTR, pRF4] was generated from our previous project (*Otarigho and Aballay, 2021*). The strains were crossed with the WT laboratory N2. All strains were grown at 20°C on nematode growth medium (NGM) plates seeded with *E. coli* OP50 as the food source (*Brenner, 1974*) unless otherwise indicated. The recipe for the control NGM plates is: 3 g/l NaCl, 3 g/l peptone, 20 g/l agar, 5 µg/ml cholesterol, 1 mM MgSO₄, 1 mM CaCl₂, and 25 mM potassium phosphate buffer (pH 6.0). The NGM plates were without antibiotics except as indicated.

## RNA interference

Knockdown of targeted genes was obtained using RNAi by feeding the animal with *E. coli* strain HT115(DE3) expressing double-stranded RNA homologous to a target gene (*Fraser et al., 2000*; *Timmons and Fire, 1998*). RNAi was carried out as described previously (*Sun et al., 2011*). Briefly, *E. coli* with the appropriate vectors were grown in LB broth containing ampicillin (100 µg/ml) and tetracycline (12.5 µg/ml) at 37°C overnight and plated onto NGM plates containing 100 µg/ml ampicillin and 3 mM isopropyl β-D-thiogalactoside (RNAi plates). RNAi-expressing bacteria were grown at 37°C for 12–14 hr. Gravid adults were transferred to RNAi-expressing bacterial lawns and allowed to lay eggs for 2–3 hr. The gravid adults were removed, and the eggs were allowed to develop at 20°C to young adults. This was repeated for another generation (except for ELT-2 RNAi) before the animals were used in the experiments. The RNAi clones were from the Ahringer RNAi library.

## *C. elegans* survival assay on bacterial pathogens

*P. aeruginosa* and *S. enterica* were incubated in LB medium. *S. aureus* was incubated in a TSA medium with nalidixic acid (20 µg/ml). The incubations were done at 37°C with gentle shaking for 12 hr. *P. aeruginosa* and *S. enterica* were grown on a modified NGM agar medium of 0.35% peptone and TSA, respectively. For partial lawn assays, 20 µl of the overnight bacterial cultures were seeded at the center of the relevant agar plates without spreading. For full-lawn experiments, 20 µl of the bacterial culture was seeded and spread all over the surface of the agar plate. No antibiotic was used for *P. aeruginosa* and *S. enterica*, while nalidixic acid (20 µg/ml) was used for the TSA plates for *S. aureus*. The seeded plates were allowed to grow for 12 hr at 37°C. The plates were left at room temperature for at least 1 hr before the infection experiments. 20 synchronized young adult animals were transferred to the plates for infection, three technical replicate plates were set up for each condition (n=60 animals), and the experiments were performed in triplicate. The plates were then incubated at 25°C. Scoring was performed every 12 hr for *P. aeruginosa* and *S. aureus*, and 24 hr for *S. enterica*. Animals were scored as dead if the animals did not respond to touch by a worm pick or lack of pharyngeal pumping. Live animals were transferred to fresh pathogen lawns each day. All *C. elegans* killing assays were performed three times independently.

## Bacterial lawn avoidance assay

Bacterial lawn avoidance assays were performed by 20 ml of *P. aeruginosa* PA14 and *S. aureus* NCTCB325 on 3.5 cm modified NGM agar plates (0.35% peptone) and (0.35% TSA) respectively, which were cultured at 37°C overnight to have a partial lawn. The modified NGM plates were left to cool to room temperature for about 1 hr, and 20 young adult animals grown on *E. coli* OP50 were transferred to the center of each bacterial lawn afterward. The number of animals on the bacterial lawns was counted at 12 and 24 hr after exposure.

## Aversive training

Training plates of 3.5 cm diameter containing either *E. coli* OP50 on SK agar or *S. aureus* on TSA agar were prepared as described previously. Young gravid adult hermaphroditic animals that were grown on *E. coli* OP50 were washed with M9 and transferred to the training plates. They were allowed to roam for 4 hr at 25°C. After this, the animals were rewashed and transferred to a TSA plate containing lawn occupancy TSA-plated seed with *S. aureus*, as described above. The number of animals on the bacterial lawns was counted.

## Avoidance assays at 8% oxygen

Avoidance assays as described above were carried out in a hypoxia chamber. Briefly, after young gravid adult hermaphroditic animals were transferred to the avoidance plates, the plates were placed in the hypoxia chamber, and the lids of the plates were removed. The chamber was purged with 8% oxygen (balanced with nitrogen) for 5 min at a flow rate of 25 l/min. The chamber was then sealed, and assays were carried out. Control plates were incubated at ambient oxygen.

## Pharyngeal pumping rate assay

WT and *npr-15(tm12539)* animals were synchronized by placing 20 gravid adult worms on NGM plates seeded with *E. coli* OP50 and allowing them to lay eggs for 60 min at 20°C. The gravid adult worms were then removed, and the eggs were allowed to hatch and grow at 20°C until they reached the young adult stage. The synchronized worms were transferred to NGM plates fully seeded with *P. aeruginosa* for 24 hr at 25°C. Worms were observed under the microscope with a focus on the pharynx. The number of contractions of the pharyngeal bulb was counted over 60 s. Counting was conducted in triplicate and averaged to obtain pumping rates.

## Defecation rate assay

WT and *npr-15(tm12539)* animals were synchronized by placing 20 gravid adult worms on NGM plates seeded with *E. coli* OP50 and allowing them to lay eggs for 60 min at 20°C. The gravid adult worms were then removed, and the eggs were allowed to hatch and grow at 20°C until they reached the young adult stage. The synchronized worms were transferred to NGM plates fully seeded with *P. aeruginosa* for 24 hr at 25°C. Worms were observed under a microscope at room temperature. For each worm, an average of 10 intervals between 2 defecation cycles were measured. The defecation cycle was identified as a peristaltic contraction beginning at the posterior body of the animal and propagating to the anterior part of the animal followed by feces expulsion.

## Brood size assay

The brood size assay was done following the earlier described methods (*Berman and Kenyon, 2006*; *Otarigho and Aballay, 2020*). Ten L4 animals from egg-synchronized populations were transferred to individual NGM plates (seeded with *E. coli* OP50) (described above) and incubated at 20°C. The animals were transferred to fresh plates every 24 hr. The progenies were counted and removed every day.

## *C. elegans* longevity assays

Longevity assays were performed on NGM plates containing live, UV-killed *E. coli* strains HT115 or OP50 as described earlier (*Sun et al., 2011*; *Kumar et al., 2019*; *Otarigho and Aballay, 2021*; *Otarigho and Aballay, 2020*). Animals were scored as alive, dead, or gone each day. Animals that failed to display touch-provoked or pharyngeal movement were scored as dead. Experimental groups contained 60–100 animals and the experiments were performed in triplicate. The assays were performed at 20°C.

## Intestinal bacterial loads visualization and quantification

Intestinal bacterial loads were visualized and quantified as described earlier (*Sun et al., 2011*; *Otarigho and Aballay, 2020*). Briefly, *P. aeruginosa*-GFP lawns were prepared as described above. The plates were cooled to ambient temperature for at least an hour before seeding with young gravid adult hermaphroditic animals and the setup was placed at 25°C for 24 hr. The animals were transferred from *P. aeruginosa*-GFP plates to the center of fresh *E. coli* plates for 10 min to eliminate *P. aeruginosa*-GFP

on their body. The step was repeated two times more to further eliminate external *P. aeruginosa*-GFP left from earlier steps. Subsequently, 10 animals were collected and used for fluorescence imaging to visualize the bacterial load while another 10 were transferred into 100 µl of PBS plus 0.01% Triton X-100 and ground. Serial dilutions of the lysates ($10^1$–$10^{10}$) were seeded onto LB plates containing 50 µg/ml of kanamycin to select for *P. aeruginosa*-GFP cells and grown overnight at 37°C. Single colonies were counted the next day and represented as the number of bacterial cells or CFU per animal.

## Fluorescence imaging

Fluorescence imaging was carried out as described previously (*Otarigho and Aballay, 2020*). Briefly, animals were anesthetized using an M9 salt solution containing 50 mM sodium azide and mounted onto 2% agar pads. The animals were then visualized for bacterial load using a Leica M165 FC fluorescence stereomicroscope. The diameter of the intestinal lumen was measured using Fiji-ImageJ software. At least 10 animals were used for each condition.

## RNA sequencing and bioinformatic analyses

Approximately 40 gravid WT and *npr-15(tm12539)* animals were placed for 3 hr on 10 cm NGM plates (seeded with *E. coli* OP50) (described above) to have a synchronized population, which developed and grew to L4 larval stage at 20°C. Animals were washed off the plates with M9 and frozen in QIAzol by ethanol/dry ice and stored at –80°C prior to RNA extraction. Total RNA was extracted using the RNeasy Plus Universal Kit (QIAGEN, Netherlands). Residual genomic DNA was removed using TURBO DNase (Life Technologies, Carlsbad, CA, USA). A total of 6 µg of total RNA was reverse-transcribed with random primers using the High-Capacity cDNA Reverse Transcription Kit (Applied Biosystems, Foster City, CA, USA).

The library construction and RNA sequencing in Illumina NovaSeq 6000 platform was done following the method described by *Zhu et al., 2018*, and *Yao et al., 2018*, pair-end reads of 150 bp were obtained for subsequent data analysis. The RNA sequence data were analyzed using a workflow constructed for Galaxy (https://usegalaxy.org) as described (*Jalili et al., 2020*) and was validated using Lasergene DNA star software. The RNA reads were aligned to the *C. elegans* genome (WS271) using the aligner STAR. Counts were normalized for sequencing depth and RNA composition across all samples. Differential gene expression analysis was then performed on normalized samples. Genes exhibiting at least twofold change were considered differentially expressed. The differentially expressed genes were subjected SimpleMine tools from WormBase (https://www.wormbase.org/tools/mine/simplemine.cgi) to generate information such as WormBase ID and gene name, which are employed for further analyses. Gene ontology analysis was performed using the WormBase IDs in DAVID Bioinformatics Database (https://david.ncifcrf.gov) (*Dennis et al., 2003*) and validated using a *C. elegans* data enrichment analysis tool (https://wormbase.org/tools/enrichment/tea/tea.cgi). The enrichment analysis tool on WormBase indicates that all significantly enriched terms have a q value less than 0.1. Immune and age determination pathways were obtained using the Worm Exp version 1 (http://wormexp.zoologie.uni-kiel.de/wormexp/) (*Yang et al., 2016*) using the transcription factor target category. The Venn diagrams were obtained using the web tool InteractiVenn (http://www.interactivenn.net) (*Heberle et al., 2015*) and bioinformatics and evolutionary genomics tool (http://bioinformatics.psb.ugent.be/webtools/Venn/). While neuron wiring was done using the database of synaptic connectivity of *C. elegans* for computation (*White et al., 1986*, http://ims.dse.ibaraki.ac.jp/ccep-tool/).

## RNA isolation and qRT-PCR

Animals were synchronized and total RNA extraction was done following the protocol described above. Quantitative reverse transcription-PCR (qRT-PCR) was conducted using the Applied Biosystems One-Step Real-time PCR protocol using SYBR Green fluorescence (Applied Biosystems) on an Applied Biosystems 7900HT real-time PCR machine in 96-well plate format. Twenty-five microliter reactions were analyzed as outlined by the manufacturer (Applied Biosystems). The relative fold changes of the transcripts were calculated using the comparative $CT(2^{-\Delta\Delta CT})$ method and normalized to pan-actin (*act-1*, *–3*, *–4*). The cycle thresholds of the amplification were determined using StepOnePlus Real-Time PCR System Software v2.3 (Applied Biosystems). All samples were run in triplicate. The primer sequences were available upon request and presented in *Supplementary file 7*.

## Generation of transgenic *C. elegans*

To generate *npr-15* rescue animals, the DNA *npr-15* alongside its promoter was amplified from the genomic DNA of Bristol N2 *C. elegans* adult worms. Plasmid pPD95_77_*Pnpr-15_npr-15* was constructed by linearization of plasmid pPD95_77 using HindIII and SalI restriction digestion enzymes. The amplified *Pnpr-15::npr-15* DNA was cloned behind its native promoter in the plasmid pPD95_77, between HindIII and SalI sites. For the neuronal rescue strain, the plasmid pPD95_77_*Prab-3_npr-15* was constructed by cloning amplified *npr-15* DNA into BamHI/KpnI digested pPD95_77_*Prab-3* under the promoter of *rab-3*. The constructs were purified and sequenced. Young adult hermaphrodite *npr-15(tm12539)* and WT *C. elegans* were transformed by microinjection of plasmids into the gonad as described (**Mello et al., 1991**; **Mello and Fire, 1995**). Briefly, a mixture containing pPD95_77_*Pnpr-15_npr-15* (40 ng/µl) and *Pmyo-3*::mCherry (5 ng/µl) that drives the expression of mCherry to the muscle as a transformation marker was injected into the animals. For the neuronal rescue, a mixture containing pPD95_77_*Prab-3_npr-15* plasmids (40 ng/µl) and *Pmyo-3*::mCherry (5 ng/µl) that drives the expression of mCherry to the pharynx as a transformation marker was injected into the animals. To rescue *npr-15* in ASJ neurons, *Ptrx-1* promoter and *npr-15* were amplified from gDNA. The *npr-15* fragment was cloned downstream of the *Ptrx-1* by in-fusion PCR to obtain *Ptrx-1::npr-15*, which was cloned into pPD95.77 between the PstI and SmaI sites to generate npr-*15(tm12539);Ptrx-1::npr-15*. The ASJ neuron-specific rescue strain *15(tm12539);Ptrx-1::npr-15* was generated by injecting plasmid pPD95_77_ *Ptrx-1::npr-15* (25 ng/µl) with *Pmyo-2*::mCherry (10 ng/µl) as a co-injection marker into the *npr-15(tm12539)* and WT which were crossed to *npr-15(tm12539)* animals. Primers used in the generation of transgenic animals are shown in **Supplementary file 7**.

## Quantification and statistical analysis

Statistical analysis was performed with Prism 8 version 8.1.2 (GraphPad). All error bars represent the standard deviation. The two-sample t-test was used when needed, and the data were judged to be statistically significant when $p<0.05$. In the figures, asterisks (*) denote statistical significance as follows: NS, not significant, *, $p<0.01$, **, $p<0.001$, ***, $p<0.0001$, as compared with the appropriate controls. The Kaplan-Meier method was used to calculate the survival fractions, and statistical significance between survival curves was determined using the log-rank test. All experiments were performed at least three times.

## Acknowledgements

This work was fully supported by NIH grants GM0709077 and AI117911 (to AA). Most strains used in this study were obtained from the Caenorhabditis Genetics Center (CGC), which is funded by the NIH Office of Research Infrastructure Programs (P40 OD010440) and the National BioResource Project (NBRP) of Japan.

## Additional information

### Funding

| Funder | Grant reference number | Author |
| --- | --- | --- |
| National Institutes of Health | GM0709077 | Alejandro Aballay |
| National Institutes of Health | AI117911 | Alejandro Aballay |

The funders had no role in study design, data collection and interpretation, or the decision to submit the work for publication.

### Author contributions

Benson Otarigho, Conceptualization, Data curation, Formal analysis, Validation, Investigation, Visualization, Methodology, Writing - original draft, Writing - review and editing; Anna Frances Butts, Data curation, Formal analysis, Methodology, Writing - review and editing; Alejandro Aballay,

Conceptualization, Data curation, Supervision, Funding acquisition, Investigation, Visualization, Writing - original draft, Project administration, Writing - review and editing

**Author ORCIDs**
Benson Otarigho  http://orcid.org/0000-0001-6347-3831
Alejandro Aballay  http://orcid.org/0000-0002-5975-3352

Reviewer #1 (Public Review): https://doi.org/10.7554/eLife.90051.4.sa1
Reviewer #2 (Public Review): https://doi.org/10.7554/eLife.90051.4.sa2
Author Response https://doi.org/10.7554/eLife.90051.4.sa3

## Additional files

### Supplementary files
• Supplementary file 1. Screening G-protein-coupled receptor NPR Mutants for Unknown Immunological Roles in Immune Defense. (A) Unknown immune role NPR_GPCRs, (B) Screen of NPRs mutants that have not be known to play a role in immunity.

• Supplementary file 2. Mean, standard deviation, and standard error of lawn occupancy across different experimental trials in wild-type (WT) and *npr-15(tm12539)*.

• Supplementary file 3. Upregulated and downregulated genes in *npr-15(tm12539)* vs wild-type (WT) RNA Seq.

• Supplementary file 4. Upregulated immune genes and pathways in *npr-15(tm12539)* vs wild-type (WT) RNA Seq.

• Supplementary file 5. Gene inactivation in wild-type (WT) and exposed to *S. aureus* infection for aversion behavior.

• Supplementary file 6. Upregulated neuropeptide genes in *npr-15(tm12539)* vs wild-type (WT) RNA Seq.

• Supplementary file 7. Primers used in this study.

• MDAR checklist

### Data availability
Sequencing data have been deposited in GEO under accession code GSE173367.

The following dataset was generated:

| Author(s) | Year | Dataset title | Dataset URL | Database and Identifier |
|---|---|---|---|---|
| Otarigho B, Aballay A | 2024 | RNA Sequencing of npr-15(tm12539) and wild type (WT) N2 C elegans | http://www.ncbi.nlm.nih.gov/geo/query/acc.cgi?acc=GSE173367 | NCBI Gene Expression Omnibus, GSE173367 |

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
