## [Editor Report · eLife assessment]

The **important** work by Aballay et al. significantly advances our understanding of how G protein-coupled receptors (GPCRs) regulate immunity and pathogen avoidance. The authors provide **convincing** evidence for the GPCR NPR-15 to mediate immunity by altering the activity of several key transcription factors. This work will be of broad interest to immunologists.

---

## [Referee Report · Reviewer #1 (Public Review)]

Summary:

Otarigho et al. presented a convincing study revealing that in *C. elegans*, the neuropeptide Y receptor GPCR/NPR-15 mediates both molecular and behavioral immune responses to pathogen attack. Previously, three npr genes were found to be involved in worm defense. In this study, the authors screened mutants in the remaining npr genes against *P. aeruginosa*-mediated killing and found that npr-15 loss-of-function improved worm survival. npr-15 mutants also exhibited enhanced resistance to other pathogenic bacteria but displayed significantly reduced avoidance to S. aureus, independent of aerotaxis, pathogen intake and defecation. The enhanced resistance in npr-15 mutant worms was attributed to upregulation of immune and neuropeptide genes, many of which were controlled by the transcription factors ELT-2 and HLH-30. The authors found that NPR-15 regulates avoidance behavior via the TRPM gene, GON-2, which has a known role in modulating avoidance behavior through the intestine. The authors further showed that both NPR-15-dependent immune and behavioral responses to pathogen attack were mediated by the NPR-15-expressing neurons ASJ. Overall, the authors discovered that the NPR-15/ASJ neural circuit may regulate distinct defense mechanisms against pathogens under different circumstances. This study provides novel and useful information to researchers in the fields of neuroimmunology and *C. elegans* research.

Strengths:

1. This study uncovered specific molecules and neuronal cells that regulate both molecular immune defense and behavior defense against pathogen attack and indicate that the same neural circuit may regulate distinct defense mechanisms under different circumstances. This discovery is significant because it not only reveals regulatory mechanisms of different defense strategies but also suggests how *C. elegans* utilize its limited neural resources to accomplish complex regulatory tasks.

2. The conclusions in this study are supported by solid evidence, which are often derived from multiple approaches and/or experiments. Multiple pathogenic bacteria were tested to examine the effect of NPR-15 loss-of-function on immunity; the impacts of pharyngeal pumping and defecation on bacterial accumulation were ruled out when evaluating defense; RNA-seq and qPCR were used to measure gene expression; gene inactivation was done in multiple strains to assess gene function.

3. Gene differential expression, gene ontology and pathway analyses were performed to demonstrate that NPR-15 controls immunity through regulating immune pathways.

4. Elegant approaches were employed to examine avoidance behavior (partial lawn, full lawn, and lawn occupancy) and the involvement of neurons in regulating immunity and avoidance (the use of a diverse array of mutant strains).

5. Statistical analyses were appropriate and adequate.

---

## [Referee Report · Reviewer #2 (Public Review)]

Summary:

The authors are studying the behavioral response to pathogen exposure. They and others have previously describe the role that the G-protein coupled receptors in the nervous system plays in detecting pathogens, and initiating behavioral patterns (e.g. avoidance/learned avoidance) that minimize contact. The authors study this problem in *C. elegans*, which is amenable to genetic and cellular manipulations and allow the authors to define cellular and signaling mechanisms. This paper extends the original idea to now implicate signaling and transcriptional pathways within a particular neuron (ASJ) and the gut in mediating avoidance behaviour.

Strengths:

The work is rigorous and elegant and the data are convincing. The authors make superb use of mutant strains in *C. elegans*, as well tissue specific gene inactivation and expression and genetic methods of cell ablation. to demonstrate how a gene, NPR15 controls behavioral changes in pathogen infection. The results suggest that ASJ neurons and the gut mediate such effects. I expect the paper will constitute an important contribution to our understanding of how the nervous system coordinates immune and behavioral responses to infection.

---

## [Author Response]

**eLife assessment**
The important work by Aballay et al. significantly advances our understanding of how G protein-coupled receptors (GPCRs) regulate immunity and pathogen avoidance. The authors provide convincing evidence for the GPCR NPR-15 to mediate immunity by altering the activity of several key transcription factors. This work will be of broad interest to immunologists.

The authors express their sincere appreciation to Timothy Behrens (Senior Editor), the Reviewing Editor, and the original reviewers for their considerate and favorable assessment of our manuscript.

**Reviewer #1 (Public Review):**
Summary:Otarigho et al. presented a convincing study revealing that in *C. elegans*, the neuropeptide Y receptor GPCR/NPR-15 mediates both molecular and behavioral immune responses to pathogen attack. Previously, three npr genes were found to be involved in worm defense. In this study, the authors screened mutants in the remaining npr genes against *P. aeruginosa*-mediated killing and found that npr-15 loss-of-function improved worm survival. npr-15 mutants also exhibited enhanced resistance to other pathogenic bacteria but displayed significantly reduced avoidance to S. aureus, independent of aerotaxis, pathogen intake and defecation. The enhanced resistance in npr-15 mutant worms was attributed to upregulation of immune and neuropeptide genes, many of which were controlled by the transcription factors ELT-2 and HLH-30. The authors found that NPR-15 regulates avoidance behavior via the TRPM gene, GON-2, which has a known role in modulating avoidance behavior through the intestine. The authors further showed that both NPR-15-dependent immune and behavioral responses to pathogen attack were mediated by the NPR-15-expressing neurons ASJ. Overall, the authors discovered that the NPR-15/ASJ neural circuit may regulate distinct defense mechanisms against pathogens under different circumstances. This study provides novel and useful information to researchers in the fields of neuroimmunology and *C. elegans* research.

The authors are grateful for the thoughtful and insightful comments on our manuscript. Your feedback has been instrumental in refining our work, and we appreciate the time and expertise you have invested in evaluating our study.

Strengths:1. This study uncovered specific molecules and neuronal cells that regulate both molecular immune defense and behavior defense against pathogen attack and indicate that the same neural circuit may regulate distinct defense mechanisms under different circumstances. This discovery is significant because it not only reveals regulatory mechanisms of different defense strategies but also suggests how *C. elegans* utilize its limited neural resources to accomplish complex regulatory tasks.

The authors express gratitude to the reviewer for recognizing that the present study revealed specific molecules and neuronal cells involved in regulating both molecular immune defense and behavioral defense against pathogen attacks. Additionally, the acknowledgment that the same neural circuit may oversee distinct defense mechanisms under different circumstances is appreciated.

1. The conclusions in this study are supported by solid evidence, which are often derived from multiple approaches and/or experiments. Multiple pathogenic bacteria were tested to examine the effect of NPR-15 loss-of-function on immunity; the impacts of pharyngeal pumping and defecation on bacterial accumulation were ruled out when evaluating defense; RNA-seq and qPCR were used to measure gene expression; gene inactivation was done in multiple strains to assess gene function.

The authors thank the reviewer for appreciating that this study is supported by solid evidence.

1. Gene differential expression, gene ontology, and pathway analyses were performed to demonstrate that NPR-15 controls immunity by regulating immune pathways.

The authors thank the reviewer for appreciating the Gene differential expression, gene ontology, and pathway analyses performed in the study.

1. Elegant approaches were employed to examine avoidance behavior (partial lawn, full lawn, and lawn occupancy) and the involvement of neurons in regulating immunity and avoidance (the use of a diverse array of mutant strains).

The author thanks the reviewer for appreciating the approaches used in this study.

1. Statistical analyses were appropriate and adequate.

The authors thank the reviewer for appreciating the Statistical analyses used in this study.

**Reviewer #2 (Public Review):**
Summary:The authors are studying the behavioral response to pathogen exposure. They and others have previously describe the role that the G-protein coupled receptors in the nervous system plays in detecting pathogens, and initiating behavioral patterns (e.g. avoidance/learned avoidance) that minimize contact. The authors study this problem in *C. elegans*, which is amenable to genetic and cellular manipulations and allow the authors to define cellular and signaling mechanisms. This paper extends the original idea to now implicate signaling and transcriptional pathways within a particular neuron (ASJ) and the gut in mediating avoidance behaviour.Strengths:The work is rigorous and elegant and the data are convincing. The authors make superb use of mutant strains in *C. elegans*, as well tissue specific gene inactivation and expression and genetic methods of cell ablation. to demonstrate how a gene, NPR15 controls behavioral changes in pathogen infection. The results suggest that ASJ neurons and the gut mediate such effects. I expect the paper will constitute an important contribution to our understanding of how the nervous system coordinates immune and behavioral responses to infection.

The authors sincerely thank the reviewer for the thoughtful and positive review of our manuscript. We greatly appreciate the time and effort you dedicated to evaluating our work, and we are pleased that you find our study to be a rigorous and elegant contribution to the understanding of behavioral responses to pathogen exposure.

**Reviewer #1 (Recommendations For The Authors):**
The authors have adequately addressed my concerns and questions. I have no more comments or recommendations for the authors.

The authors thank the reviewer for the constructive comments on the manuscript

**Reviewer #2 (Recommendations For The Authors):**
The authors have adequately addressed my concerns.

The authors express their appreciation to the reviewer for the valuable and constructive comments provided on the manuscript.